# Functionalization of Poly(styrene-co-methyl methacrylate) Particles for Selective Removal of Bilirubin

**DOI:** 10.3390/ma15175989

**Published:** 2022-08-30

**Authors:** María del Prado Garrido, Ana Maria Borreguero, Francisco Javier Redondo, David Padilla, Manuel Carmona, María Jesús Ramos, Juan Francisco Rodriguez

**Affiliations:** 1Department of Chemical Engineering, Institute of Chemical and Environmental Technology, University of Castilla-La Mancha, Avda. De Camilo José Cela 1, 13005 Ciudad Real, Spain; 2Department of Anesthesiology and Critical Care Medicine, University General Hospital, Obispo Rafael Torija s/n, 13005 Ciudad Real, Spain; 3Faculty of Medicine, University of Castilla-La Mancha, Camino de Moledores s/n, 13005 Ciudad Real, Spain; 4Department of Surgery, University General Hospital, Obispo Rafael Torija s/n, 13005 Ciudad Real, Spain

**Keywords:** poly(ethylene glycol) methacrylate, glycidyl-methacrylate, hyperbilirubinemia, albumin immobilization, suspension polymerization, wetting effect

## Abstract

Hyperbilirubinemia is one of the main causes of death in patients with severe hepatic problems, which justifies the research for bilirubin removal solutions. In this study, St-MMA particles with PEGMA and/or GMA brushes were synthesized. First, the recipe for St-MMA was optimized and then adapted for PEGMA and GMA incorporation. Different solvents were then assayed to improve the BSA immobilization capacity of the particles. Ethyl lactate proved to be the best solvent, reaching a BSA immobilization capacity improvement of up to 60% for St-MMA-GMA-PEGMA particles. These particles also presented the best results for BR removal from PBS. No significant differences in the final capacity for BR removal from PBS media were observed when BSA was attached to the particles; however, the kinetics were greatly improved, requiring half the time. Finally, St-MMA-GMA-PEGMA particles that were wetted in EL with BSA reduced the bilirubin concentration in plasma from levels that threaten the survival of critical patients to levels close to those of healthy individuals in less than 30 min. On the contrary, particles without BSA were unable to remove bilirubin from plasma. Thus, the attachment of albumin to the particles plays a key role in selectively reducing bilirubin levels.

## 1. Introduction

Bilirubin (BR) is the yellow breakdown product of normal heme catabolism resulting from the body’s removal of aged red blood cells, which contain hemoglobin. It is a toxic and water-insoluble product that requires an adequate transport mechanism in the body for evacuation. Bilirubin can be found in the human body in three states: direct or conjugated bilirubin, which is linked to glucuronic acid and is stored in the gall bladder for its later excretion; non-conjugated bilirubin, which is mostly linked to albumin while transported by the bloodstream; and free bilirubin, which is the non-conjugated bilirubin that is neither linked to albumin nor to glucuronic acid. Conjugated bilirubin can be filtered by a healthy kidney in case of liver failure, whereas the non-conjugated bilirubin can not. Free bilirubin values in blood can dramatically increase (above the normal limit of 1.2 mg/dL) due to the consumption of certain drugs or as a result of some liver illnesses, causing hyperbilirubinemia. This illness causes death in 80% of patients with severe hepatic problems [1]. One of the reasons for this is its accumulation in tissues, especially in the brain membranes, causing *kernicterus* [2].

The problem of a bilirubin metabolism malfunction is especially severe in dialyzed patients since the dialysis membrane is refractory to the bilirubin. Indeed, the survival expectation for hemodialyzed patients with simultaneous hepatic failure is very low. Currently, the solution for bilirubin metabolism malfunction is treatment with special devices based on the usage of large volumes of concentrated albumin solutions; the MARS^®^ kit incorporated in Prismaflex (Baxter) devices is the only one that is commercially available.

With about one million people dying annually from kidney failure globally, more than two million patients in the world receive some type of dialysis; approximately 73,000 dialysis machines are being used worldwide, and growing. The necessity of developing a breakthrough bilirubin removal system is clearly justified.

There have been different approaches used in the development of bilirubin-capturing materials [3,4,5,6,7]. Thus far, materials based on polystyrene, polystyrene-divinylbenzene [8], anion exchange resins [9], activated carbon, cellulose matrix combined with cationic binding polymers [10] and even bile acid inhibitors based on sodium salts [11], have been described to remove bilirubin. Unfortunately, these systems have generally shown low selectivity and capacity [12]. The immobilization of albumin onto the surface of blood compatible particles can be a good alternative to solve this problem, since this approach would imitate the mechanism of bilirubin capture in the body, hence increasing its selectivity.

Protein immobilization is a process in which a protein is retained on a support in such a way that its biological activity is mostly maintained or even enhanced. Protein immobilization has some advantages, such as increases in protein stability and the ability to reuse the material, thereby reducing the costs; however, it also has some disadvantages such as modification of the original state of the protein, which can reduce to some extent that protein’s activity [13,14].

Nakanishi et al. proposed an adsorption process of bovine serum albumin (BSA) over polystyrene, polyvinylidene fluoride or nitrocellulose membranes [15]; however, the weakness of the interaction between the protein and the substrates made the protein easily lixiviated. Another immobilization path is the protein’s impregnation or encapsulation into a solid structure where it is trapped in the polymeric network of a gel, or inside microcapsules; however, diffusion of the metabolite (in this case, bilirubin) through the capsule or the gel is greatly hindered [16].

Some of the approaches to albumin non-site-specific immobilization onto a support have been carried out on membranes and flat surfaces. Bergström and Holmberg linked albumin to polystyrene plates containing polyethylene glycol (PEG) chains as functional brushes that are able to interact through hydrophilic and hydrogen bond interactions with the albumin. The laying of the albumin onto the PEG brushes makes the interactions between the albumin and the rigid support softer, thus preserving the albumin’s natural conformation. Furthermore, the presence of PEG on the surface prevents surface bio-fouling [17]. Albumin has also been attached to polypropylene membranes coated with polyethylene glycol methacrylate brushes, where the polyethylene glycol methacrylate improves the material’s biocompatibility [18]. However, these aforementioned supports have disadvantages, such as an adsorption capacity for the protein that is limited to 300 ng/cm^2^ [19]. An agarose–tungsten carbide composite adsorbent with an anionic mixed-mode ligand has been studied by Gomes et al. [20] to immobilize albumin in order to employ such composite particles as novel multimodal adsorbents in expanded bed adsorption (EBA) chromatographic separations; however, this approach yields adsorption capacities commonly lower than 450 ηg of albumin per gram of solid support.

Covalent bond immobilization involves linkage of the protein to a reactive group of the solid support by means of a covalent link, in such a way that this does not affect the conformation of the protein, and maintains or even enhances its biological activity [21,22]. Specifically, materials based on polystyrene coated with polypyrrole [23], polyaniline nanowires (PANI-NWs) [24] and flat sheet membranes from polyvinylidene fluoride (PVDF) [25] and from functionalized polytetrafluoroethylene (PTFE) [26] have also been investigated for BSA immobilization, with promising results.

Nevertheless, in order to achieve a high capture velocity in an adsorption process, abundant and readily available active centers are required, which can be limited in the flat geometries of membranes and plates as a result of their small transfer areas per mass unit. Thus, albumin immobilization onto particles as a result of their greater surface per mass unit seems more interesting. Moreover, porous particle materials also have internal linking sites available. However, the problem with this is that large internal surfaces are usually associated with small pore sizes, where the diffusion of large molecules, such as proteins, is limited [27]. For instance, for physiologic pHs, the pore diffusivity of albumin into an agarose–tungsten carbide composite was reported to be very low, with values that do not exceed 3 × 10^6^ cm^2^·min^−1^ [20]. One solution to this problem is the use of wet particulate material, which increases the pore size of the particles. In addition, the use of wet particles increases the material weight, facilitating the ability of the material to remain inside the solution, thus improving the mass transfer between them [28,29].

Another path to facilitate easier, faster and more effective bilirubin uptake while avoiding mass transfer problems would be the anchoring of as the maximum possible albumin onto a particle’s external surface, paying less attention to what is allocated in the inner part of the microparticle’s polymeric backbone. Brush polymerization involves the creation of a bristle-like layer of polymer chains on the surfaces of membranes or particles, which can reduce protein attaching resistance [30]. These brushes must be inert to molecules which should not be linked, flexible enough to prevent bio-fouling of the material and present no hydrophobic interactions. Poly(2-hydroxyethyl methacrylate) (PHEMA) has been studied to create a layer of these polymer brushes for non-site-specific immobilization of albumin [31]. Diethylene glycol dimethacrylate (DEGMA) and 4-vinylpiridyne brushes have been previously used as brushes in silica particles, improving protein immobilization and reducing bio-fouling of the material [32]. Starch has also been proven to create brushes onto PMMA surfaces, increasing the capacity for the immobilization of L-asparaginase and enhancing enzyme activity in terms of thermal and pH stabilities [33]. Poly(N-methacryloyl-l-leucine) (PNML) brushes have been also used for selective albumin immobilization, showing that protein immobilization onto these filaments does not depend on temperature and pH [34]. Poly(ethylene glycol) methacrylate (PEGMA) also achieves the aforementioned characteristics, improving blood compatibility of most of materials [18] and demonstrating good flexibility for interactions with proteins [35]. In fact, PEGMA has been previously used for albumin immobilization onto surface functionalized polymer microparticles based on poly(styrene-co-methyl methacrylate) [36].

However, the incorporation of PEGMA hampers the synthesis of particulate material due to its tendency to coagulate [37], making necessary further research into the obtention of particulate materials with an accurate size suitable for these kinds of applications.

An alternative tool described in literature for albumin attachment involves material functionalization with glycidyl-methacrylate (GMA) [31,33,35,38]. The GMA presents an epoxide group that can be opened by the nucleophilic amino groups of the protein which easily react with it, covalently [14,36,37,39,40,41,42,43].

Considering these developments, the use of mixed brush polymers from PEGMA and GMA can represent an improvement in albumin immobilization, combining PEGMA hemocompatibility with the epoxide groups from GMA that facilitate covalent albumin immobilization [14,18]. In this research, both monomers were incorporated into St-MMA particles in order to study the influence of their presence in BSA immobilization. Moreover, in order to improve BSA access to the linking sites and increase the material capacity for BSA immobilization, the particles were wetted with different organic solvents. The wetting of the particles has been demonstrated to be a novel method for increasing a material’s capacity for BSA attachment. Finally, the effectiveness and selectivity of the best developed particles to remove bilirubin was tested by using not only phosphate buffer solution as a medium, as most of works do, but also human plasma.

## 2. Materials and Methods

### 2.1. Materials

The monomers methyl methacrylate (MMA, 99 wt%, Sigma-Aldrich Chemical Co. (St. Louis, MI, USA)) and styrene (St, 99 wt%, Sigma-Aldrich Chemical Co.) were purified by washing with an aqueous sodium hydroxide solution (1.25 N) and using calcium chloride as a desiccant. The monomers PEGMA (Mn 360, Sigma-Aldrich Chemical Co.) and GMA (Sigma-Aldrich Chemical Co.) were passed through neutral aluminum oxide (99 wt%, Alfa Aesar) in order to remove the inhibitor before use. The remaining agents were used as received, without further purification. Benzoyl peroxide (BPO, Luperox A75^®^, Sigma-Aldrich Chemical Co) was used as an initiator. Polyvinylpyrrolidone (PVP, K90, Mw 360000, Fluka) was used as a suspension agent. Nitrogen of high-purity grade was used to work under an inert atmosphere. BSA (Heat Shock Fraction, ≥96 wt%) and bilirubin (≥95 wt%) were supplied by Sigma-Aldrich Chemical Co. Phosphate-buffered saline (PBS, 0.01 M phosphate buffer, pH 7.4), used to immobilize the BSA and bilirubin removal experiments, was prepared freshly. Sodium chloride (NaCl, ≥99%, Sigma), potassium chloride (KCl, ACS, Panreac), di-sodium hydrogen phosphate 12-hydrate (Na_2_HPO_4_·12H_2_O, Pharma Grade, Panreac) and potassium di-hydrogen phosphate (KH_2_PO_4_, for Analysis, Panreac) were used for PBS synthesis. Sodium carbonate (Na_2_CO_3_, ≥99.5%, Sigma-Aldrich), sodium hydroxide (NaOH, 98%, Panreac), copper sulphate pentahydrate (CuSO_4_·5H_2_O, 99.995%, Sigma-Aldrich), Folin–Ciocalteu’s Reagent (Supelco) and sodium tartrate dihydrate (C_4_H_4_Na_2_O_6_·2H_2_O, ≥99.5%, Sigma) were used for protein determination using the Lowry method. For the wetting of particles, glycerol (≥99.5% Sigma-Aldrich), dimethyl sulfoxide (DMSO, ≥99.9%, Sigma-Aldrich), diethylene glycol (DEG, 99.8%, Campi y Jové S.A), ethanol (EtOH, 96°, Guinama) and ethyl lactate (EL, ≥98%, Sigma-Aldrich) were used without further purification. A standard Cromatest (Linear) kit was used to measure bilirubin in plasma. Water with a conductivity of 1 μS/cm was produced in our laboratory using distillation followed by deionization using ion exchange. Human plasma was supplied by the blood bank of University General Hospital of Ciudad Real.

### 2.2. Characterization

#### 2.2.1. Scanning Electron Microscopy

The morphology and the surface features of the synthesized particles were observed by using a Quanta 250 scanning electron microscope (FEI Company, Hillsboro, OR, USA) with a tungsten filament that operated at a working potential of 12.5 or 15 kV. All of the SEM micrographs are shown with a 500-micrometer scale.

#### 2.2.2. Particle Size and Polydispersity Index

Volume median particle size (dv_0.5_) and the polydispersity index (PdI) of the particles were determined by low-angle laser light scattering (LALLS), utilizing a Malvern Mastersizer 2000 equipped with a Scirocco 2000 unit for analyzing particles dispersed in air, and the software Mie Theory was used to analyze the experimental data.

#### 2.2.3. Copolymer Composition

The compositions of the different developed materials were studied using FT-IR. Infrared spectra were obtained using a Spectrum Two spectrometer (Perkins Elmer, Inc., Waltham, MA, USA) equipped with a universal attenuated total reflectance (UATR) accessory. The samples were scanned from 450 to 4000 cm^−1^ at room temperature.

#### 2.2.4. Swelling Percentage

The swelling percentage (%Sw) was determined, leaving the microparticles in contact with ethyl lactate for 8 h at 25 °C. The swelling percentage was measured as the difference in weight between the initial weight of particles and the final weight of swollen particles. Equation (1) was applied to determine the swelling degree, as follows:(1)$$\%\mathrm{Sw}=\frac{W_{0}-W_{f}}{W_{0}}$$
where W_0_ is the initial weight of the microparticles (g) and W_f_ is the final weight of swollen microparticles (g).

#### 2.2.5. Epoxide Content

The particles’ epoxide content was determined through a standard titration method UNE-EN ISO 3001 [44].

First, a solution of the particles in 10 mL of chloroform, 20 mL of glacial acetic acid and 10 mL of a bromide tetraethylammonium solution was prepared. Then, the mixture was titrated with a standard solution of perchloric acid (0.1 M) in acetic acid. Finally, Equation (2) was applied to calculate the epoxide content or Epoxide Index (EI) in mmol/g.
(2)$$\mathrm{EI}=\frac{\left(V_{1}-V_{0}\right)\cdot \left(1-\frac{t-t_{s}}{1000}\cdot c\right)}{1000\cdot m}$$
where m is the sample weight (g), V_1_ and V_0_ are the volumes (mL) of perchloric acid solution used in the titrations of the sample and the blank, respectively; t and t_s_ (°C) are the temperatures of the perchloric acid solution during the titration of the blank and the sample, respectively, and c is the concentration of the perchloric solution (0.1 mol/L).

#### 2.2.6. Protein Concentration Determination

The Lowry method [45] was used to determine and quantify the amount of protein that was immobilized on the microparticles. Briefly, 400 μL of BSA solution per sample was mixed with 2 mL of copper sulphate solution, and after ten minutes, Folin–Ciocalteu’s reagent was added to the sample, reacting for 30 min at room temperature. The absorbance was measured at 750 nm using a UV-VIS apparatus (JASCO V-750) that was provided with the software Spectra Manager. Phosphate buffer solution (0.01 M) was used as blank, and BSA was used as the reference standard in the protein estimation, using a linear range of 0.025–0.5 mg/mL. All of the measurements were performed in duplicate.

#### 2.2.7. Bilirubin Concentration Determination

Bilirubin in PBS was measured directly at 450 nm in a UV-VIS apparatus (JASCO V-750). Phosphate buffer solution (0.01 M) was used as blank and bilirubin (BR) was used as the reference standard in the bilirubin estimation, using a linear range of 0–3 mg/dL. All of the measurements were performed in duplicate.

Bilirubin in plasma (total and direct) was measured with a Cromatest (Linear) kit in a UV-VIS apparatus. For total bilirubin, 100 μL of plasma with bilirubin were mixed with 1 mL of working reagent, and after two minutes, the absorbance was measured at 540 nm. For direct bilirubin, 100 μL of plasma were mixed with 1 mL of direct reagent, measuring the absorbance five minutes later at 540 nm. Indirect bilirubin was measured as the difference between total and direct bilirubin.

The biochemical analysis of plasma was measured in the biochemical laboratory located at University General Hospital, in Ciudad Real (HGUCR), in a Beckman Coulter Analyzer.

### 2.3. Synthesis and Experimental Procedures

#### 2.3.1. Synthesis of St-MMA based Particles

Suspension polymerization reactions were performed in a 2-L jacketed glass reactor equipped with a reflux condenser, a nitrogen gas inlet tube, a digitally controlled stirrer, and a thermostatic bath to keep the reaction at the required conditions.

The synthesis included two phases: a continuous phase, containing water and the suspending agent; and a discontinuous one, containing the polymer monomers and benzoyl peroxide. Styrene, methyl methacrylate, poly(ethylene glycol methacrylate) and glycidyl-methacrylate monomers were used, depending on the particle composition that was desired. First, different mass ratios of monomers:water (0.15, 0.22 and 0.29) were studied for the St-MMA suspension polymerization in order to optimize the reaction yield and particle size. Then, the ratio of monomers:water was fixed at 0.29 for the synthesis of functionalized particles with PEGMA and/or GMA. The compositions of the different performed reactions are shown in Table 1 and Table 2.

Experiments were carried out following the method described in the bibliography [46]. First, water and PVP were charged into the reactor by fixing the agitation at 400 rpm and heating until 60 °C was reached. After that, the discontinuous phase was added into the continuous phase, considering this point as zero time. The initiator was dissolved and premixed with styrene and methyl methacrylate at room temperature, avoiding idle time. In the case of St-MMA particles, the polymerization process was carried out at 80 °C for 2 h, at 90 °C for the next 2 h, and at 100 °C for the last 3 h. In the case of functionalized particles with glycidyl methacrylate and/or poly(ethylene glycol) methacrylate, all of the monomers were charged together to the reactor from the beginning, and the reaction was carried out at 80 °C for 6 h. The percentage of functional monomers included in the microparticles recipe is in the same range as those reported in literature about surface functionalization for protein immobilization [33,38]. A schematic of the synthesis of functionalized particles with GMA and PEGMA is shown in Figure 1.

Once obtained, the particles were purified through repeated washing with ethanol and filtration. Finally, the product was dried at room temperature for 72 h.

#### 2.3.2. Albumin Immobilization

Albumin immobilization was carried out in a USP 4 flow-through cell apparatus (Erweka). A solution of BSA in PBS was prepared from protein and PBS at room temperature. The concentration of BSA solution was 0.2 mg BSA/mL in PBS. The solution was transferred to seven glass bottles with magnetic stirring. An amount of 2.5 g of particles was added to the particle cells, and the solution of BSA was driven through the particles bed for 72 h with a flow rate of 8 mL/min. The experiments were carried out at 25 °C. Once the BSA was attached to the particles, the samples were left to dry at 40 °C for 96 h. The installation of the USP 4 flow-through cell apparatus is shown in Figure 2.

For experiments with wetted particles, the particles were kept in contact with the corresponding solvent (glycerol, diethylene glycol, dimethyl sulfoxide, ethanol or ethyl lactate) for 20 min prior to their use for albumin immobilization. Then, the excess solvent was removed, and the wetted particles were introduced into the column of the shallow bed device.

Both types of experiments were carried out three times under the same conditions, with the results showing the average of the three experiments. A schematic that illustrates albumin immobilization to an epoxide group is shown in Figure 3.

#### 2.3.3. Bilirubin Removal Tests

Experiments to investigate the efficiency of synthesized particles, with and without BSA, for bilirubin removal from both PBS and plasma, were performed. The use of plasma allowed us to also study the selectivity towards BR compared to the rest of the molecules presented in human blood. Both kinds of experiments were carried out while maintaining the installation set up in the dark to avoid bilirubin degradation.

For experiments with PBS, different amounts of bilirubin were dissolved in 100 mL of sodium hydroxide (0.06 M) and added to 300 mL of PBS at room temperature. Between 1.5 and 2.0 g of the particles with or without BSA attached were introduced into a shallow bed device with a diameter of 3 cm and a length of 4 cm. The bilirubin solution was forced to circulate through the particles bed for 2 h with a flow rate of 120 mL/min. The experiment was carried out at room temperature.

For studying the selectivity and capacity in closer to real conditions, different amounts of bilirubin were dissolved in 100 mL of sodium hydroxide (0.06 M) and added to human plasma and stirred for thirty minutes, adjusting the pH to 7.4. Then, it was forced to circulate through the particles bed (using the same weight of particles and device as those used for the PBS experiments) for a maximum 1.5 h with a flow rate of 120 mL/min. The experiments were carried out at room temperature.

For experiments involving wetted particles, the particles were kept in contact with ethanol, methanol or ethyl lactate for 10 min prior to their use for bilirubin removal. Then, the excess solvent was removed, and the wetted particles were introduced into the column of the shallow bed device.

Both types of experiments were carried out three times under the same conditions, with the results showing the average value of the three experiments.

## 3. Results

### 3.1. St-MMA Particles Synthesis

The first step was to adjust the polymerization recipe for increasing the reaction yield, and obtaining particles of St and MMA with a proper size in order to allow the flow of human blood through the particles bed without causing coagulation or large pressure drops, while allowing fast mass transfer. For this purpose, three different ratios between the water and monomers with values of 0.15, 0.22 and 0.29 were used. The monomers to BPO weight ratio was fixed at 74 for all the experiments.

All of the used ratios allowed us to achieve spherical particles of micron size, as can be observed in their SEM micrographs shown in Figure 4.

SEM micrographs show that the concentration of monomers and initiator does not affect significantly the shape of the particles, hence spherical and smooth particles in the micron size range for all experiments were obtained. However, the particle size seems to be affected by the particular monomer to water mass ratio, increasing with the monomer concentration. This influence was confirmed by the LALLS measurements. The median diameters in volume (Dpv 0.5) and polydispersity indexes (PdI) are shown in Table 3, which also includes the reaction yield.

The LALLS results confirmed that the higher the monomers to water ratio, the higher the particle size, which can reduce the available surface for BSA attachment. Thus, higher monomers to water ratios were not assayed, since the value of 0.29 allowed us to achieve a good polymerization yield (85.2%) while keeping a low particle size in the range of those materials applicable in hemoperfusion cartridges [47]. At this point, it is important to remark that a small particle size can compromise the application of the material in hemoperfusion cartridges, since it can provoke large pressure drops that could cause coagulation of the blood. The following Ergun equation Equation (3) was applied to determine the pressure drop caused by the developed particles:(3)$$\frac{\Delta P}{L}=150\cdot V\cdot \mu \cdot \frac{1}{D_{p}^{2}}\cdot \frac{\left(1-\varepsilon \right)}{\varepsilon^{3}}+1.75\cdot \frac{V^{2}\cdot \rho }{\mathrm{Dp}}\cdot \frac{\left(1-\varepsilon \right)}{\varepsilon^{3}}$$
where ΔP(Pa) is the pressure drop of the particles bed; L (m) is the length of the particles bed; V (m/s) is the speed of the fluid through the particles bed, which is commonly 1.5 × 10^−4^; μ (kg/m·s) is the viscosity of the fluid, with a value of 4.5 × 10^−3^ for blood; Dp (m) is the particle size; ε is the porosity of the particles bed; and ρ (kg/m^3^) is the density of the fluid, which has a value of 1050 for blood.

Thus, considering the median particle size of 722 μm for the synthesized particles with a monomer to water mass ratio of 0.29 and a void fraction of the particles bed of 0.35, the theoretical pressure drop per meter of bed is 14 mmHg/m. Assuming that the length will be similar to that of a standard hemodialysis cartridge, which does not exceed 250 mm, the estimated pressure drop in the bed would be 4 mmHg. This value is quite a bit lower than the 200 mmHg allowed for maximum pressure drops for extracorporeal systems in order to avoid blood coagulation [48]. That means that the synthesized particles can be incorporated into standard hemodialysis cartridges without causing blood coagulation problems, and the 0.29 monomer to water mass ratio was selected for further functionalization studies involving PEGMA and/or GMA.

### 3.2. PEGMA- and/or GMA-Coated Particles Synthesis

As alluded to in the introduction, the incorporation of PEGMA to increase the hemocompatibility of the material and GMA as a linker moiety for albumin immobilization to the polymeric backbone of the particles is one of the goals of this research. The polymerization recipes for this study are shown in Table 2.

The successful incorporation of both monomers was confirmed using FT-IR spectroscopy (Figure 5) and by the epoxy groups determination for the case of PEGMA and GMA, respectively.

St-MMA particles with and without functionalization showed an absorption peak at 1735 cm^−1^ due to the carbonyl groups of PMMA, and at 735 cm^−1^ due to the benzene ring deformation vibration of PSt. A slight broad peak appears in the range of 3500 to 3350 cm^−1^ for the particles functionalized with PEGMA due to the presence of hydroxyl groups, confirming PEGMA incorporation. It was not possible to confirm the functionalization of the particles with GMA by means of FT-IR because of the overlay of the epoxide ring peak (915 cm^−1^) with the characteristic absorption vibration of PMMA (987 cm^−1^). Thus, in order to confirm GMA incorporation, the epoxide content in the particles was measured.

The epoxide content of the particles functionalized with GMA was determined according to the UNE-EN-ISO 3001. The epoxide content was 0.21 mmol/g particle for only the GMA functionalized case, and 0.12 mmol/g particle in the case of GMA and PEGMA functionalization.

As commented before, the particle size is a key property of these materials since the flow of human blood must be allowed through them when placed in a hemodialysis cartridge; coagulation or large pressure drops must not result, while presenting high surface for mass transfer. The median diameter in volume (Dp_v 0.5_), polydispersity index (PdI) and the estimated pressure drop that will result when blood circulates through a cartridge full of the particles are shown in Table 4. The pressure drops were estimated by the Ergun equation, considering the same assumptions described for the pressure drop calculation for St-MMA particles.

The results show an increase in the particle size when GMA is added to the P(St-co-MMA), with a Dp_v 0.5_ 32.3% higher, while PEGMA promoted smaller particles (Dp_v 0.5_ 20.5% smaller) even in combination with GMA (Dp_v 0.5_ 53.7% smaller). Despite the effects of particle size, all the developed particles are in the range of those applicable in hemoperfusion cartridges [47]. Moreover, the pressure drops estimated by the Ergun equation are quite a bit lower than the maximum allowed for their incorporation into standard hemodialysis cartridges without causing blood coagulation problems [48].

Then, the morphology of the different synthesized particles was analyzed using SEM (Figure 6).

All the particles are quite similar with respect to their morphologies and external surfaces, independent of their composition. As observed in the LALLS results (Table 4), the particle size from the St-MMA-GMA-PEGMA reaction is clearly smaller than the others, although the stirring and the surfactant concentration were the same in all the cases. This reduction in the particle size has been previously observed when PEGMA is added for the synthesis of polymeric particles [49,50].

Finally, the swelling percentage of the particles in ethyl lactate was also determined. St-MMA-GMA showed the lowest value with 133%, and St-MMA had a value of 152%. PEGMA and GMA-PEGMA functionalized particles had swelling percentages of 639 and 1209%, respectively. The use of GMA implied the lowest swelling percentage due to its use as slight cross-linker [51]. In the case of PEGMA, its higher hydrophilicity allowed it to retain a higher amount of solvent [52] that causes an expansion in the matrix and improves the access of the protein to inner epoxide groups in the particle.

### 3.3. BSA Immobilization

Albumin is a protein which contains some amino acids that allow its covalent immobilization to PEGMA- and/or GMA-brushed particles. In order to carry out this process, pH control around 7.4 (the physiological one) is required to avoid denaturing of the protein [18].

The amount of BSA retained in the particles was determined as the difference between the initial and final BSA concentrations in PBS solution, as measured by ultraviolet spectroscopy following the procedure described in the characterization section. The evolution of BSA concentrations in PBS with time is shown in Figure 7.

As can be seen in Figure 5, both monomers, the PEGMA and the GMA, increase the poly(styrene-co-methyl methacrylate) capacity for BSA immobilization. The particles containing the hydroxyl functionalization with PEGMA represented the best support for albumin immobilization, with an average value of immobilized albumin of 2.6 ± 0.69 mg BSA/g of particles. The best behavior of the particles having only hydroxyl groups functionalized can be attributed to the presence of more long-length filaments, which improves the access of large molecules such the BSA to the functional group [49]. The St-MMA-GMA and St-MMA-GMA-PEGMA particles immobilized 1.0 ± 0.27 and 1.26 ± 0.15 mg BSA/g of particles, respectively, while the value for particles without functionalization was just 0.6 mg BSA/g. Thus, the presence of only GMA also improves BSA linking, thanks to the covalent groups; however, this capacity is improved by the PEGMA addition, thanks again to the presence of filament brushes with hydroxyl groups. This way, albumin attachment to the epoxide group of glycidyl methacrylate is favored by the damping effect that the flexible chain and the OH group of the poly(ethylene glycol) methacrylate can provide. Furthermore, the brush structure of the particle surface could make more possible the attachment with the albumin conformation and its further activity, compared to its direct adsorption onto the St-MMA particle surface.

Despite the improvement in the ability of the particles to immobilize BSA that was achieved through functionalization, it was somewhat limited. Thus, the wetting effect of several solvents on the capability of St-MMA-GMA-PEGMA particles for albumin immobilization was investigated (Figure 8). St-MMA-GMA-PEGMA was chosen to study the wetting effect, since they have both functionalities.

In Figure 8 are shown the kinetics of albumin immobilization onto non-wetted particles as well as those wetted with diethylene glycol, glycerol, dimethyl sulfoxide and ethyl lactate. Ethanol was discarded due to its interference with the protein determination process of the Lowry method [53].

There were evident differences in the immobilization capacity between wetted and dry particles. In the case of glycerol, no significant improvement was observed with respect to the non-wetted particles. On the contrary, diethylene glycol, dimethyl sulfoxide and ethyl lactate presented significant improvements; this can be explained by the swelling of the particles that provides better access to the linking sites for BSA attachment inside the particles [29]. The polymer swelling significantly changers the polymer matrix volume, and moves it to the rubbery domain [54] in which higher permeabilities are observed [55]. Particles wetted with diethylene glycol and dimethyl sulfoxide presented a capacity to immobilize albumin about three times more than dry particles. Finally, ethyl lactate was the best solvent option for improving the albumin immobilization capacity of St-MMA-GMA-PEGMA particles, progressing from 1.26 ± 0.15 mg BSA/g of particles to 3.2 ± 0.38 mg BSA/g of particles, which is a 60% improvement.

The kinetics of BSA immobilization are also enhanced, since the maximum uptakes are reached in just 24, 16 and 8 h for the cases or using diethylene glycol, dimethyl sulfoxide and ethyl lactate, respectively; these are much shorter times than those necessary for dry particles, which did not become saturated even after 60 h (Figure 7).

Taking into account these results (the fastest and greatest improvements in BSA immobilization), ethyl lactate was the solvent selected for albumin immobilization with the rest of the particles for further BR removal tests.

The albumin immobilization capacities studied for the rest of the particles once wetted with EL were 2.0 ± 0.36 and 4.0 ± 0.04 mg BSA/g of particles for St-MMA and St-MMA-PEGMA, respectively. St-MMA-GMA particles became agglomerated when kept in contact with ethyl lactate, impeding their use for bilirubin removal.

### 3.4. Bilirubin Removal Tests

The main target of this research is to effectively remove bilirubin present in human blood, in order to improve the recovery expectations of patients with hyperbilirubinemia. As explained in the synthesis and experimental procedures section, PBS and plasma were used as media for the experiments to investigate the efficiency of synthesized particles for bilirubin removal.

First, PBS was used to study the kinetics of bilirubin uptake by the particles without and with BSA wetted in EL. The kinetics of bilirubin removal from PBS with time are shown in Figure 9.

According to these results, the BR removal capacity is much faster in the presence of BSA, since the equilibria are reached after just 40 min (Figure 9b); meanwhile, particles without BSA require about 90 min (more than double the time). However, for these ratios between BR and BSA amounts, there were no significant improvements in the final amount of bilirubin removed from the PBS, per gram of particle (Table 5).

Finally, the capacity of the particles to remove bilirubin was tested in a close-to-real medium, human plasma. The initial BR concentration was set to 8mg/dL, which is a value that can be commonly found in analytical results from people with severe hepatic failure, according to University General Hospital from Ciudad Real. It must be taken into account that, while the bilirubin used in PBS removal is in free form, in the case of human plasma there are also conjugated or direct bilirubin (dBR) as well as non-conjugated or indirect bilirubin (iBR) forms present; thus, the initial concentration will always be higher in human blood than in PBS. With this type of experiment, not only will bilirubin removal capacity be studied under more realistic conditions, but also the selectivity for the uptake of free bilirubin (which is the most harmful bilirubin) against the rest of the substances present in human plasma can be assessed. For this test, St-MMA-GMA-PEGMA particles previously wetted in ethyl lactate were selected since they presented the fastest and highest BR removal rates from PBS. Moreover, to confirm that the BR removal is due to the presence of BSA, experiments under the same conditions were also carried out with wetted St-MMA-GMA-PEGMA particles without BSA.

Figure 10 shows the kinetics of bilirubin removal from plasma by wetted St-MMA-GMA-PEGMA particles, with and without BSA.

Figure 10a shows two experiments of bilirubin removal in plasma with similar concentrations. The iBR removed by the particles in each case was 14.2 and 14.8 mg BR/g of particles for the first and second experiment, respectively, removing on average 14.5 ± 0.42 mg bilirubin/g particles in a short period of time (shorter than 30 min).

Although the upper reference value for indirect bilirubin is 0.8 mg/dL, the reached concentrations of about 1.7 mg/dL represent an acceptable value for critical patients with hyperbilirubinemia.

On the other hand, according to the results in Figure 10b, particles without BSA were not able to uptake bilirubin; thus, bilirubin removal in plasma media is related to the presence of albumin. This is probably due to the preferential uptake of other substances that are present in blood by the particles instead of BR, demonstrating a lack of selectivity towards BR of particles without BSA.

In order to complete the study, general analytical biochemistries of the plasma at the beginning and at the end of experiments of bilirubin removal using particles previously wetted in ethyl lactate, with and without BSA, were conducted (Table 6).

These experiments confirmed the selectivity of St-MMA-GMA-PEGMA particles with BSA towards indirect bilirubin (iBR), since just this metabolite and, as a consequence the total bilirubin, suffered an important change. No differences in direct bilirubin (dBR) or any other parameter from the general biochemical analyses were observed. In the case of St-MMA-GMA-PEGMA particles without BSA, the lack of BR removal capacity in plasma media was confirmed, justifying the interest in BSA-facilitated immobilization.

## 4. Conclusions

A ratio of monomers to water of 0.29 was found to be the best of the studied mixtures for St-MMA particles synthesis, since it obtained the highest reaction yield (85.2%), while promoting a particle size (722 µm) that is suitable for their application in conventional hemoperfusion cartridges, as well as a low polydispersity (PdI = 0.705). Then, with this monomer to water ratio, St-MMA particles functionalized with PEGMA and/or GMA brushes were successfully synthesized, with particle sizes also in the range of those materials applied in conventional hemoperfusion cartridges. The theoretical expected pressure drops for blood circulation through them were also determined, with a maximum value of 70 mmHg for St-MMA-GMA-PEGMA, which was considerably lower than the limit for extracorporeal systems (200 mmHg). The wetting of the synthesized particles with diethylene glycol, DMSO and ethyl lactate improved significantly BSA immobilization capacity. Ethyl lactate was the best solvent, reaching a BSA immobilization capacity improvement of up to 60% for the case of St-MMA-GMA-PEGMA particles, with a final BSA capacity of 3.2 mg/g of particles. These particles also presented the best results for BR removal from PBS. Although there were no significant differences in the final capacity for BR removal from PBS media when BSA was attached to the particles, the kinetics of BR removal were greatly improved by the presence of BSA, requiring just half the time to reach equilibrium. Finally, St-MMA-GMA-PEGMA particles wetted in EL with BSA were applied to reduce the bilirubin concentration in plasma; it was possible to conclude that they can reduce BR levels in critical patients, from those which threaten survival to levels close to those of healthy individuals, in a time shorter than 30 min. On the contrary, particles without BSA were not able to remove bilirubin from plasma. Thus, attachment of albumin to the particles played a key role in selectively reducing bilirubin levels.

## Figures and Tables

**Figure 1 materials-15-05989-f001:**
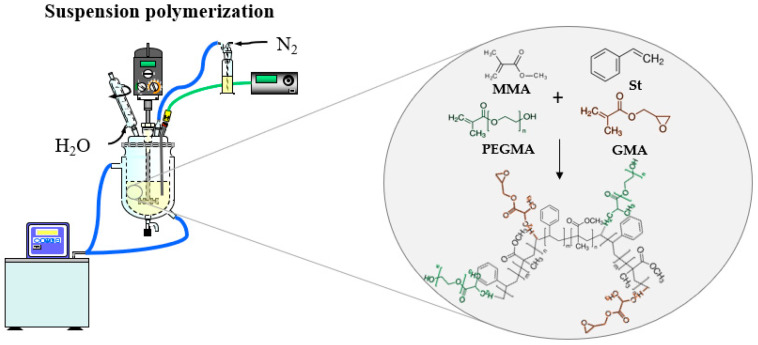
Schematic of the synthesis of functionalized particles.

**Figure 2 materials-15-05989-f002:**
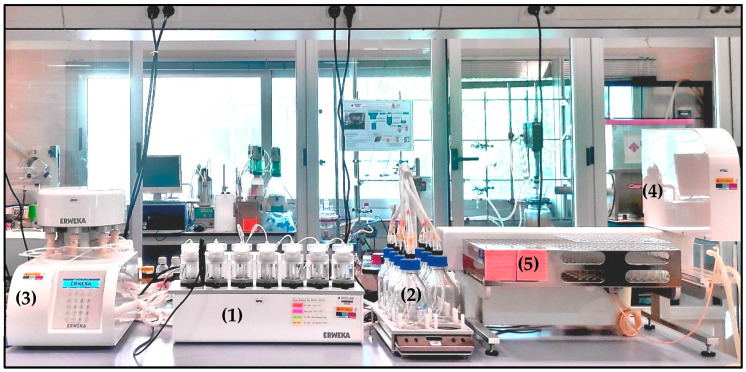
Flow-through cell apparatus (USP 4). (**1**) Seven jacketed polycarbonate particle flow-through cells, (**2**) media transfer station with seven flasks, (**3**) piston pump for the pumping of solutions through the particle cells, (**4**) peristaltic pump for the extraction of samples, (**5**) automatic sample collector.

**Figure 3 materials-15-05989-f003:**
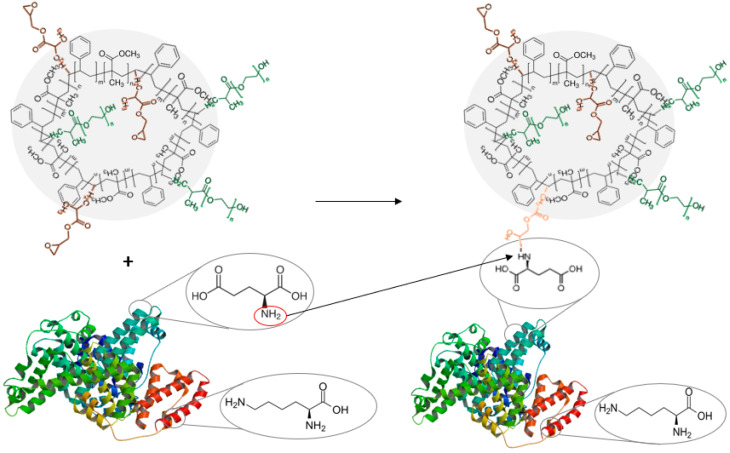
Schematic showing immobilization of BSA to an epoxide group from GMA.

**Figure 4 materials-15-05989-f004:**
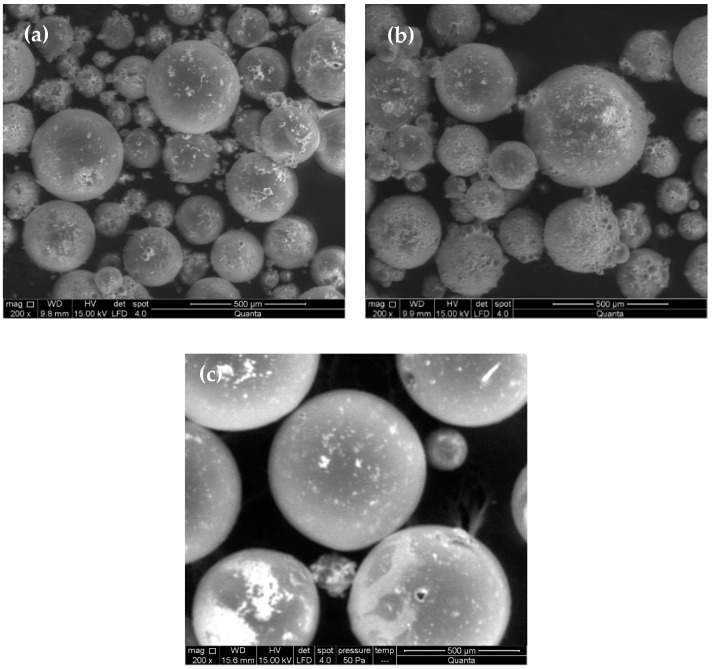
SEM micrographs of particles synthesized by suspension polymerization using different monomers to water ratios: 0.15 (**a**), 0.22 (**b**) and 0.29 (**c**).

**Figure 5 materials-15-05989-f005:**
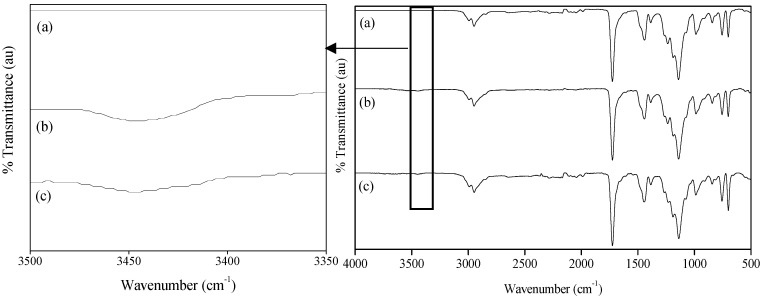
FT-IR spectra of the different synthesized particles (**right**), St-MMA (**a**), St-MMA-PEGMA particles (**b**) and St-MMA-GMA-PEGMA particles (**c**) and ampliation of FT-IR spectra of the different synthesized particles (**left**), St-MMA (**a**), St-MMA-PEGMA particles (**b**) and St-MMA-GMA-PEGMA particles (**c**), for wavenumber between 3350 and 3500 cm^−1^.

**Figure 6 materials-15-05989-f006:**
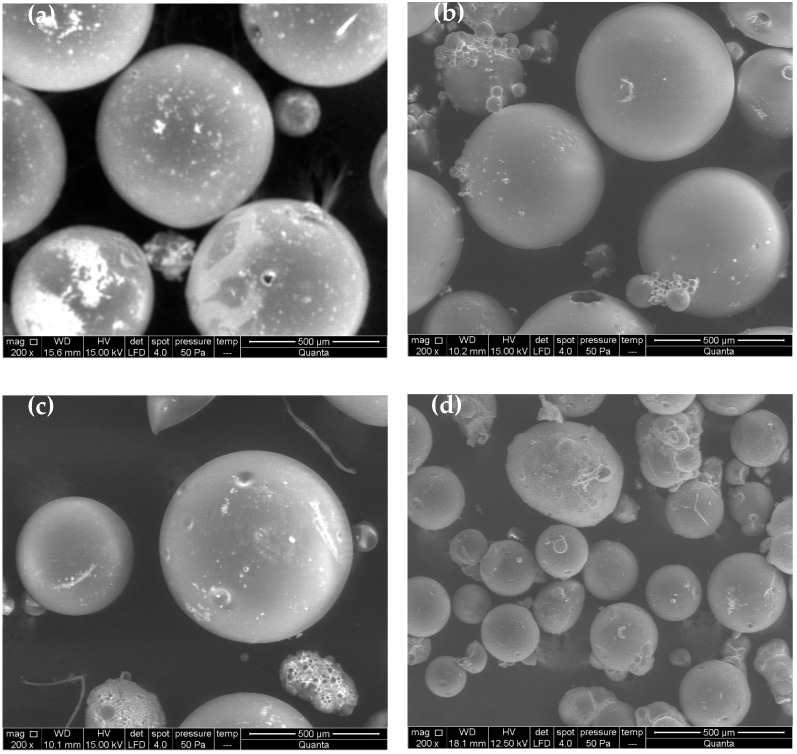
SEM micrographs of St-MMA particles (**a**), St-MMA-PEGMA particles (**b**), St-MMA-GMA particles (**c**) and St-MMA-GMA-PEGMA particles (**d**).

**Figure 7 materials-15-05989-f007:**
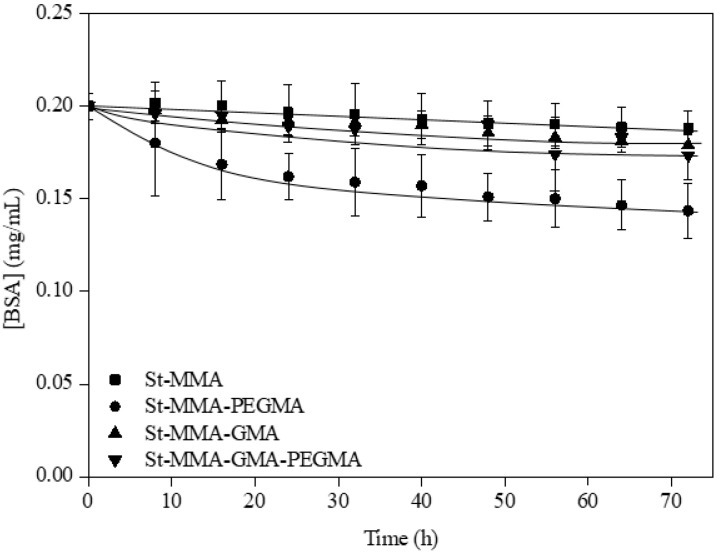
Kinetics of BSA immobilization in non-wetted particles. All of the experiments were carried out with 2.5 g of particles, at 25 °C and with a flow rate of 8 mL/min.

**Figure 8 materials-15-05989-f008:**
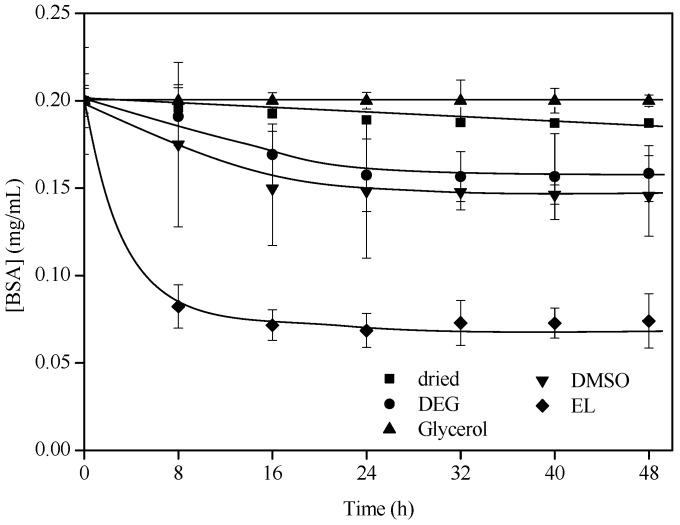
Kinetics of BSA immobilization onto St-MMA-GMA-PEGMA dry and wetted particles with different solvents. All the experiments were carried out with 2.5 g of particles, at 25 °C and with a flow rate of 8 mL/min.

**Figure 9 materials-15-05989-f009:**
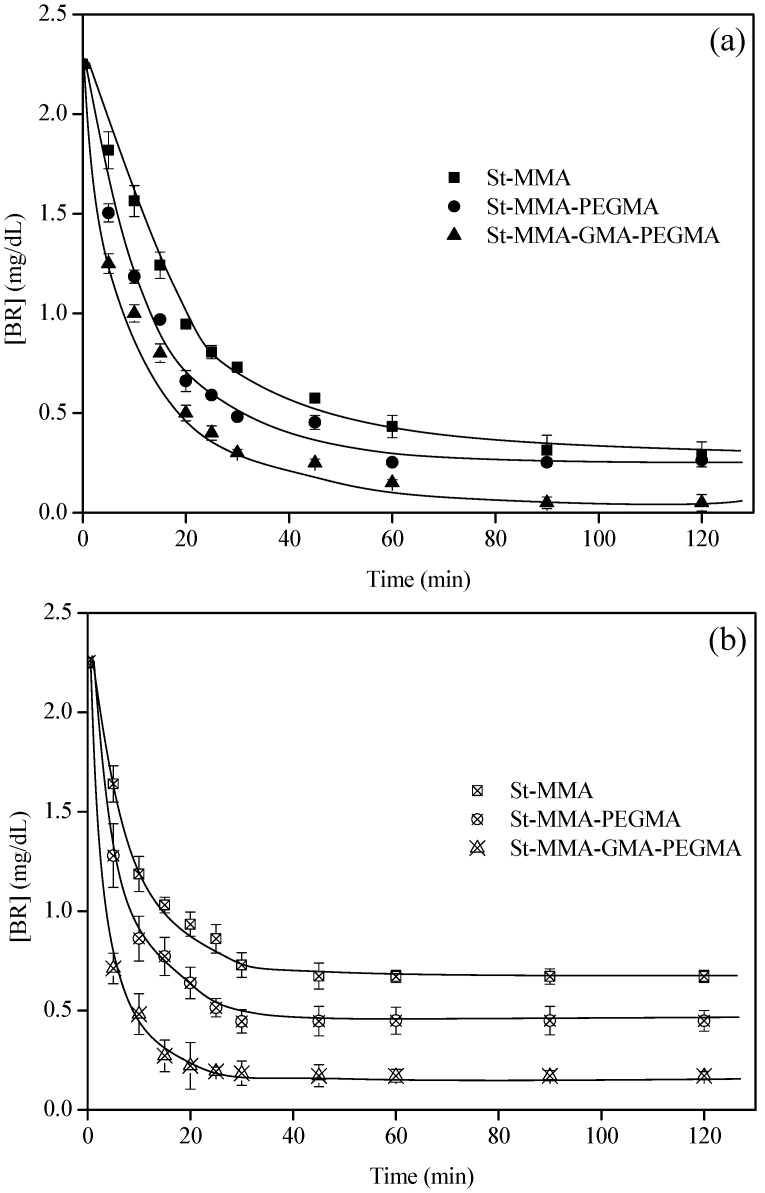
Kinetics of bilirubin removal by ethyl lactate wetted particles without (**a**) and with (**b**) BSA. The experiments were carried out with 2.0 and 1.5 g of particles, respectively. All of the experiments were conducted at 25 °C and with a flow rate of 120 mL/min.

**Figure 10 materials-15-05989-f010:**
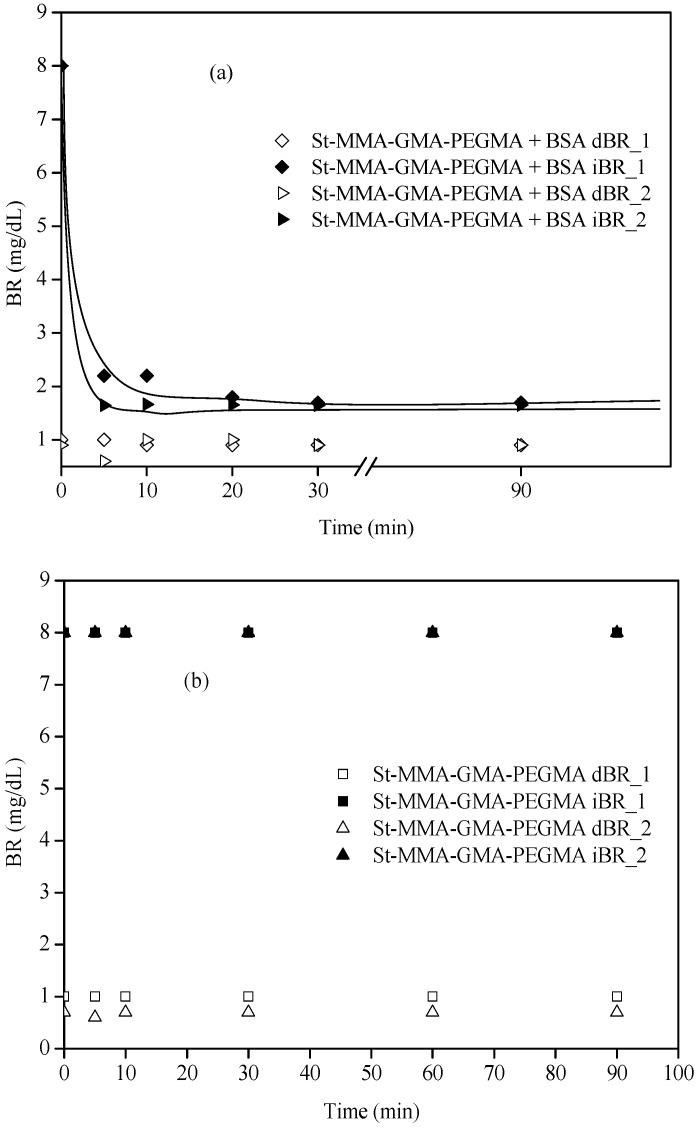
Kinetics of bilirubin removal from plasma by St-MMA-GMA-PEGMA particles with BSA (**a**) and without BSA (**b**), wetted with ethyl lactate. All of the experiments were carried out with 1.2 g of particles at 23.5 °C, and with a flow rate of 120 mL/min.

**Table 1 materials-15-05989-t001:** Reaction recipe for the optimization of the monomers to water ratio in the synthesis of St-MMA particles.

Monomers to Water Ratio	0.15	0.22	0.29
Reagent	Composition (wt%)		
**MilliQ water**	87.0	81.7	77.0
**PVP**	0.10	0.10	0.07
**BPO**	0.14	0.18	0.23
**St**	2.56	3.62	4.5
**MMA**	10.2	14.4	18.11

**Table 2 materials-15-05989-t002:** Reaction recipe for the synthesis of functionalized St-MMA-based particles.

	St-MMA-PEGMA	St-MMA-GMA	St-MMA-GMA-PEGMA
Reagent	Composition (wt%)		
**Milli-Q water**	76.2	76.2	76.5
**PVP**	0.07	0.07	0.07
**BPO**	0.23	0.23	0.23
**St**	4.5	4.5	4.2
**MMA**	18	18	18
**GMA**	-	1	0.5
**PEGMA**	1	-	0.5

**Table 3 materials-15-05989-t003:** Influence of monomers to water ratios in the polymerization yields and in the sizes and PdIs of St-MMA particles.

Experiment	Monomers to Water Ratio	Dp_v 0.5_ (μm)	PdI	Yield (%)
1	0.15	200	1.059	67.4
2	0.22	274	1.263	75.4
3	0.29	722	0.705	85.2

**Table 4 materials-15-05989-t004:** Median diameters in volume and PdIs of St-MMA functionalized particles, and estimated pressure drops.

Experiment	Dp_v 0.5_ (μm)	PdI	ΔP (mmHg)
**St-MMA-PEGMA**	574	1.114	24
**St-MMA-GMA**	955	0.693	9
**St-MMA-GMA-PEGMA**	334	0.849	70

**Table 5 materials-15-05989-t005:** Amount of removed bilirubin from PBS in particles with BSA wetted in ethyl lactate.

Particle	BR Uptake (mg/g)	
	Without BSA	With BSA
**St-MMA**	2.84 ± 0.05	2.70 ± 0.13
**St-MMA-PEGMA**	2.98 ± 0.08	3.01 ± 0.04
**St-MMA-GMA-PEGMA**	3.40 ± 0.03	3.45 ± 0.04

**Table 6 materials-15-05989-t006:** General biochemical analyses of plasma in contact with St-MMA-GMA-PEGMA particles, with and without BSA, wetted in ethyl lactate.

Reference Value	Component	Concentration (mg/dL)			
		with BSA		without BSA	
		at 0 min.	at 90 min.	at 0 min.	at 90 min.
74–106	Serum basal glucose	321	312	338	340
12.8–42.8	Serum urea	31	43	37	32
0.7–1.3 (M)/0.5–1.1 (W)	Serum creatinine	0.7	3.6	0.81	1.31
3.5–7.2 (M)/2.6–6 (W)	Serum uric acid	3.5	3.3	3.6	3.6
0.3–1.2	Serum total bilirubin	9.0	2.6	8.3	8.4
0.01–0.2	Direct bilirubin	1.0	0.9	0.9	1.0
0.1–0.8	Indirect bilirubin	8.0	1.7	7.4	7.4
6.4–8.3	Total protein	5 × 10^3^	5 × 10^3^	5.2 × 10^3^	5.2 × 10^3^
3.4–4.8	Serum albumin	3.5 × 10^3^	3.5 × 10^3^	3.7 × 10^3^	3.7 × 10^3^
<290	Serum cholesterol	129	128	136	138
50–200	Serum triglycerides	297	283	342	345
40–60	HDL-Cholesterol	16	13	28	28
<100	Direct LDL Cholesterol	74	71	75	69

## Data Availability

Contact via email the corresponding author for supporting data.

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
