# Peer review of "Functionalization of Poly(styrene-co-methyl methacrylate) Particles for Selective Removal of Bilirubin"

_materials, 2022, doi:10.3390/ma15175989_

Round 1
Reviewer 1 Report
The submitted paper reports on the synthesis of a brushes containing poly(styrene-co-methyl methacrylate). The main component investigated is St-MMA-GMA- PEGMA, in which St stands for Styrene, MMA - methyl methacrylate, GMA- glycidyl-methacrylate and PEGMA for poly(ethylene glycol) methacrylate. Authors evaluate performance of these particles for selective removal of bilirubin, which has increased concentration 9(>1.2 ng/ dL) in human blood in patients affected by hyperbilirubinemia. The manuscript seems to be well structured and written with plenty of experimental data. Still, several issues should be sorted out by the authors before the manuscript can be accepted for the publication.
My comments are:
- The text within the lines 34 -149 is repeated once again between lines 150-379. The lines 150-379 must be deleted.
- Too many unexplained abbreviation are used in the abstract such as St-MMA , PEGMA ,GMA, BR, EL, St-MMA-GMA- PEGMA. Then, within the manuscript text there is explained the meaning of abbreviations for several times. For instance, the abbreviation of PEGMA is explained in the Introduction (line 127), Materials (line 385), Results (line 580). The signification of the abbreviation should be made a single time. Also, the significance of EL in not explained within the manuscript.
- The chemical formulae should be written correctly. For instance, the Na2HPO4•12H2O (line 396) should became Na2HPO4•12H2O. The same should be applied to KH2PO4 (line 397), Na2CO3 (line 397), CuSO4•5H2O (line 399), C4H4Na2O6•2H2O (line 401).
- The author stated several times that “All of the measurements were performed in duplicate” (lines 443 and 448) both for BSA and bilirubin determinations. The measurements must be performed in triplicate in order to have a minimum evaluation of the errors.
- For the phrase “The concentration of BSA solution was 0.2 mg BSA/mL PBS” (line 493). There is a “in” missing between 0.2 mg BSA/mL and PBS.
- In Figure 3 the FTIR spectra of St-MMA, St-MMA-PEGMA and St-MMA-GMA-PEGMA particles could be overplayed in a single graphic in order to see clear the differences between the three samples.
- In graphs 7 and 8 related to bilirubin remove kinetics with St-MMA, St-MMA-PEGMA and St-MMA-GMA-PEGMA in the absence of protein as well as of St-MMA-GMA-PEGMAin the presence of BSA, please, include the errors.
Author Response
Thanks in advance for the time and effort in evaluating our manuscript. We have incorporated your valuable suggestions into our manuscript and you can see our detailed reply below.
- “The text within the lines 34 -149 is repeated once again between lines 150-379. The lines 150-379 must be deleted.”
The duplicate part of the text has been removed.
- “Too many unexplained abbreviations are used in the abstract such as St-MMA, PEGMA, GMA, BR, EL, St-MMA-GMA- PEGMA. Then, within the manuscript text there is explained the meaning of abbreviations for several times. For instance, the abbreviation of PEGMA is explained in the Introduction (line 127), Materials (line 385), Results (line 580). The signification of the abbreviation should be made a single time. Also, the significance of EL in not explained within the manuscript.”
All duplicate explanations of the abbreviations have been removed, being all the abbreviations explained once in introduction or materials and methods sections. The signification of EL is explained in line 178.
- “The chemical formulae should be written correctly. For instance, the Na2HPO4•12H2O (line 396) should became Na2HPO4•12H2O. The same should be applied to KH2PO4 (line 397), Na2CO3 (line 397), CuSO4•5H2O (line 399), C4H4Na2O6•2H2O (line 401)”.
All the commented mistakes have been corrected in materials and methods section: Line 170 for Na2HPO4•12H2O, line 171 for KH2PO4, line 172 for Na2CO3, line 173 for CuSO4•5H2O and line 174 for C4H4Na2O6•2H2O.
- “The author stated several times that “All of the measurements were performed in duplicate” (lines 443 and 448) both for BSA and bilirubin determinations. The measurements must be performed in triplicate in order to have a minimum evaluation of the errors.”
It is true that a triplicate measure offers a more accurate evaluation of the error. In general, measurement were carried out according to the standard Lowry procedure in which two replicates are generally accepted.
- “For the phrase “The concentration of BSA solution was 0.2 mg BSA/mL PBS” (line 493). There is a “in” missing between 0.2 mg BSA/mL and PBS.”
The preposition “in” has been included in line 279.
- “In Figure 3 the FTIR spectra of St-MMA, St-MMA-PEGMA and St-MMA-GMA-PEGMA particles could be overplayed in a single graphic in order to see clear the differences between the three samples”
To see better the difference due to the incorporation of the hydroxyl group in St-MMA-PEGMA and St-MMA-GMA-PEGMA, an ampliation in the range between 3500 and 3350 cm-1 has been incorporated beside to the previous graphic (before Figure 3 is now Figure 5).
- “In graphs 7 and 8 related to bilirubin remove kinetics with St-MMA, St-MMA-PEGMA and St-MMA-GMA-PEGMA in the absence of protein as well as of St-MMA-GMA-PEGMA in the presence of BSA, please, include the errors.”
The error in Figure 9 (before Figure 7) are included in the graphic and the error in the bilirubin removal kinetic are also shown in Table 5.
The error in Figure 10 (before Figure 8) are not included because in those experiments could not be done replicates. The error in the bilirubin removal is included in line 538 next to the average value of removed bilirubin.
Reviewer 2 Report
In the paper "Functionalization of Poly(styrene-co-methyl methacrylate) particles for Selective Removal of bilirubin" are presented new and interesting results about a new approach to the fabrication and application of functionalized poly(styrene-co-methyl methacrylate) particles for selective removal of bilirubin. The paper can be without a doubt accepted in Materials mdpi after major revision. A lot of the important issues should be clarified.
Please add a scheme to illustrate the process of the functionalization of the particles.
It is unclear to me why wetted and dry particles have different immobilization capacities. Authors guess that it "can be explained by the swelling of the particles that can provide better access to the linking sites for the BSA attachment inside the particle" but in this place will be valuable to add information on solubility parameters or to compare the solvents and polymer in another manner. In any case, the appropriate discussion should be added.
It is well known the structure of the BSA includes three domains. If I understood rightly, BSA molecules are immobilized onto the particles in a chaotic manner. What is the impact of the BSA orientation on bilirubin removal from PBS solution or plasma?
Please add relevant citations where similar results were presented.
https://doi.org/10.1016/j.colsurfb.2014.03.049
https://doi.org/10.1016/j.apsusc.2022.154201
Author Response
Thanks in advance for the time and effort in evaluating our manuscript. We have incorporated your valuable suggestions into our manuscript and can see our detailed reply below.
“Please add a scheme to illustrate the process of the functionalization of the particles.”
A scheme of the synthesis of functionalized particles with GMA and PEGMA has been included in Figure 1.
“It is unclear to me why wetted and dry particles have different immobilization capacities. Authors guess that it "can be explained by the swelling of the particles that can provide better access to the linking sites for the BSA attachment inside the particle" but in this place will be valuable to add information on solubility parameters or to compare the solvents and polymer in another manner. In any case, the appropriate discussion should be added.”
As described in section 2.2.1 from Materials and Methods, PEGMA and GMA are included in the particles by suspension like polymerization, being both fed to the reactor at the beginning of the reaction with styrene and methyl methacrylate. It is impossible to control if OH or epoxide groups are on the particle surface or inside the matrix, what could reduce their accessibility. An explanation has been included in Lines 475-476.
It is clearly described in literature that polymer particles has to be properly prewetted with an appropriate bridge solvent that makes more compatible and available the inner part of the particle allowing the disentanglement of the polymer chain and the access of the adsorbates in a greater portion of the particle bead.
For the sake of clarity, he swelling percentages for each type of particle were determined and included in the manuscript (the description of the calculation is described in Lines 199-206 while the results are included in Lines 419-425).
“It is well known the structure of the BSA includes three domains. If I understood rightly, BSA molecules are immobilized onto the particles in a chaotic manner. What is the impact of the BSA orientation on bilirubin removal from PBS solution or plasma?”
With chaotic manner we meant that there is not a target aminoacid involved in the process. In fact, there are several of them with amino goups that could allow the linkage with the epoxide.
It is well known that albumin has at least two sites for the link of bilirubin: one with high affinity and the most important (with an affinity constant of 1.4· 107 L/mol a 37º C), while the other sites imply a weak link. However, the albumin needs only one aminoacid to react with an epoxide group in the particle and for that reason, the sites for bilirubin removal are available an no impact is expected, independently on the media in which bilirubin is present.
“Please add relevant citations where similar results were presented.
https://doi.org/10.1016/j.colsurfb.2014.03.049
https://doi.org/10.1016/j.apsusc.2022.154201”
Both references have been included in the introduction of the article (Lines 127 and 132 respectively).
Reviewer 3 Report
Paper entitled Functionalization of Poly(styrene-co-methyl methacrylate) particles for Selective Removal of bilirubin presents a importance topic. The experimental work and manuscript are well-structured and logically built. The introduction is great and provide enough information about the background of this study. Basically, it’s a good work, but some issue should be considered before acceptance:
- In Table 1 and Figure 2: meaning of monomer ratio should be clarified, it was molar or mass ratio?
- A representative figure or scheme about the polymerization process and the function (how the BSA can be immobilized with the particles) of polymer particles should be added to the MS.
- Table 3: How many repetition were the Experiment 1, 2 and 3 made ? Details about the reproducibility of results (standard error) should be added. It’s really true for Figure 5, 6 and 7 which shows results about BSA immobilization.
- The exact meaning of abbreviations (e.g. St-MMA) should be given at each tables and figures.
- Figure 3 need to be revised, while the differences in IR spectra of sample a) b) and c) is hard to see at present form. Maybe one figure which show the 3 spectra with different color or the marking of relevant peaks can be better.
- Details of SEM images should be more visible (e.q. the scale).
Author Response
Thanks in advance for the time and effort in evaluating our manuscript. We have incorporated your valuable suggestions into our manuscript and can see our detailed reply below.
- “In Table 1 and Figure 2: meaning of monomer ratio should be clarified, it was molar or mass ratio?”
The monomer water ratio is referred to a mass ratio of the amount of styrene and methyl methacrylate divided by the amount of water. This has been clarified in section 2.2.1. “Synthesis of St-MMA based particles”, including the word “mass” for this ratio.
- “A representative figure or scheme about the polymerization process and the function (how the BSA can be immobilized with the particles) of polymer particles should be added to the MS.”
A scheme of the monomers polymerization and of the reaction between the GMA incorporated onto the particles and an aminoacid of the albumin have been included in Figures 1 and 3, respectively.
- “Table 3: How many repetitions were the Experiment 1, 2 and 3 made? Details about the reproducibility of results (standard error) should be added. It’s really true for Figure 5, 6 and 7 which shows results about BSA immobilization.”
Experiments 1, 2 and 3 were carried out once because the synthesis of St-MMA particles was previously developed in another work of the research group. These experiments were carried out just to confirm the viability of the synthesis of these particles for different monomers:water mass ratios.
Respect to the standard error for Figures 5, 6 and 7 (current Figures 7, 8, 9), errors are incorporated in the graphics and the error respect to the amount of albumin immobilized or bilirubin removed are described in the text for Figures 7 and 8 (pages 13 and 14, respectively) and in Table 5 for Figure 9.
- “The exact meaning of abbreviations (e.g. St-MMA) should be given at each tables and figures.”
Despite we agree it could be useful, we prefer to follow the Author guidelines. According to the author guidelines, the abbreviations should appear only once (Acronyms/Abbreviations/Initialisms should be defined the first time they appear in each of three sections: the abstract; the main text; the first figure or table). All the abbreviations are explained in the Introduction or Material and Methods sections.
- “Figure 3 need to be revised, while the differences in IR spectra of sample a) b) and c) is hard to see at present form. Maybe one figure which show the 3 spectra with different color, or the marking of relevant peaks can be better.”
To see better the difference due to the incorporation of the hydroxyl group in St-MMA-PEGMA and St-MMA-GMA-PEGMA, an ampliation in the range between 3500 and 3350 cm-1 has been incorporated close to the previous graphic (previous Figure 3 is now Figure 5).
- “Details of SEM images should be more visible (e.q. the scale).”
All the SEM images have been expanded to improve the visibility of the scale. In addition, the scale used in all the photos (500 µm) has also been included in Scanning Electron Microscopy paragraph in section 2.2 of Materials and Methods.
Round 2
Reviewer 2 Report
The authors have addressed all of my comments and the manuscript can be accepted for publication.
Author Response
Thanks again to the reviewer for the revison of our paper. English has been revised again.
Reviewer 3 Report
Authors replied the questions and made changes according to the comments , thus the manuscript could be published at Materials.
Author Response
Thanks again to the reviewer for the revision